# Effects of Kinesio Taping with Squat Exercise on the Muscle Activity, Muscle Strength, Muscle Tension, and Dynamic Stability of Softball Players in the Lower Extremities: A Randomized Controlled Study

**DOI:** 10.3390/ijerph19010276

**Published:** 2021-12-27

**Authors:** Hyeon-Hee Kim, Kyung-Hun Kim

**Affiliations:** 1Department of Physical Therapy, Gimcheon University, Gimcheon-si 39528, Korea; banchang0@hanmail.net; 2Gimcheon Institute of Rehabilitation Science, Gimcheon University, Gimcheon-si 39528, Korea

**Keywords:** knee joint, Kinesio taping, squat exercise, softball plays, lower extremities

## Abstract

Background and purpose: Knee injuries are common among female softball players, and the stability of the lower extremities and the strength of the knee are essential factors for them. The purpose of this study was to investigate the effect of Kinesio taping with squat exercise (KTSE) on lower extremity muscle activity, muscle strength, muscle tone, and dynamic stability of softball players. Methods: In this study, 40 softball players were randomly assigned to the KTSE group and sham taping with squat exercise (SKTSE) group. All subjects were tested three times a week for 6 weeks, i.e., for a total of 18 times. To evaluate the lower-extremity muscle activity, muscle strength, and muscle tone of the lower extremities, as well as dynamic stability, we used Noraxon Mini DTS, a digital muscular meter from JTech Medical, MyotonPRO, and the side hop test (a clinical evaluation method), respectively. These items were measured before the experiment and 6 weeks after the start of the experiment. Results: Both groups showed significant differences in lower-extremity muscle activity, muscle strength, muscle tone, and dynamic stability (*p* < 0.05). After the experiment, significant effects on lower-extremity muscle activity, muscle strength, muscle tone, and dynamic stability were observed in the KTSE group compared with in the SKTSE group (*p* < 0.05). Conclusions: KTSE did not have a negative effect on all items of the functional performance test. KTSE improved lower-extremity muscle activity, muscle strength, muscle tone, and dynamic stability.

## 1. Introduction

The ability to throw a ball accurately and quickly is essential for softball players. In the ball-throwing exercise chain, the stability of the lower extremities plays an important role in reducing the risk of injury while providing the basis for making movements; however, the injury rate of the lower extremities can be as high as 51.1% [1]. In particular, inaccurate alignment of the lower extremities, weakening of the femur, and excessive pressure can cause lesions in the knee joint. Female athletes have smaller femoral muscle tissue than their male counterparts, making them more prone to injuries [2]. Maintaining static and dynamic balance is essential for accurate posture adjustment [3].

Players must strengthen the structures of the knee joints, such as muscles, ligaments, and tendons, to prevent injuries caused by external forces, with squat exercises (SE) being one of the best knee joint exercises. Squat movements are also crucial for running and jumping, and are the most important and basic movements of the lower body as they strengthen the hip muscles, quadriceps muscle, and trunk muscles, as well as the bone density along with ligaments and tendons [4]. In addition, SE can improve the muscle strength of the lower extremities by supporting weight, and is a functional exercise that stimulates multi-joint movement, functional type of muscle mobilization, and a sense of proprioception, compared with an exercise that does not support weight [5]. As such, muscle activation of the lower extremities quickly responds to external loads and plays a role in dynamic stability such as posture and joints [6].

There are various methods for recovering muscle fatigue and activating muscles to improve muscle quality and prevent injury. Typical examples include foam rollers, muscle massages, and Kinesio taping (KT). Exercising the quadricep muscle and the back of the thigh using a foam roller has been shown to improve the flexibility of the knee joint compared with static and dynamic stretching; however, no difference has been reported in maximum muscle strength [7]. Meanwhile, the results of a study involving normal and sham massages showed that the massage was effective in relieving pain and improving the operating range after exercise, However, there was little effect in terms of quick restoration of the muscle strength [8].

KT is a natural healing process that helps to smoothen the flow of body fluid and light stimulation through the epidermis of the body, and is commonly used by many athletes for reasons including relieving a wide range of pain during exercise, such as muscle sprain and muscle tension [9]. The usefulness of KT has moved beyond protecting and fixing muscles and joints. It is now also widely for clinical purposes and in sports games. In addition, several techniques of KT are continuously developed to improve muscle function, such as muscle strength and muscle endurance [10].

In previous studies conducted on the effects of SE, squats remained a good example of closed-chain exercise, such as ankle joint bending, knee joint bending, and hip joint bending, minimizing stress on the anterior cruciate ligament [11] and increasing functional stability in patients with patellofemoral pain syndrome [12].

Previous studies of KT suggested that KT application to the shoulder bones of overhead athletes such as softball players increased the distance between shoulder bones and upper arms, proposing that it could be an effective method to improve the shoulder rotator strength and range of exercise [13]. In addition, non-athlete females performed concentric and eccentric exercises with KT, and their eccentric muscle strength improved compared with the non-KT group [14].

In this way, exercises combining KT and SE have been considered an important factor in enhancing and maintaining the lower-extremity strength in softball players. Therefore, in this study, the effect of KTSE was explored by examining the muscle activity, muscle strength, muscle tension, and dynamic stability of softball players.

## 2. Materials and Methods

### 2.1. Participants

This study used a single-blind method. The study adhered to the principles of the Declaration of Helsinki received approval from the Gimcheon University Institutional Review Board (GU-202104-HRa-04-02-P). The trial was registered under trial registration no. KCT00065800.

The inclusion criteria were as follows: (1) female softball players (age ≤ 30 years); (2) registered as a Korean softball player; (3) played in all positions except pitcher; (4) a person with >5 years of experience; (5) those without lesions in the thighs and knee joints; and (6) those who did not receive steroid or drug injections in the lower extremities within the last 30 days.

The exclusion criteria were as follows: (1) those who underwent knee and thigh-related surgery or any surgery within the last 2 years; (2) those with knee and thigh pain or lesions; (3) those with <5 years of exercise experience; and (4) those who tested positive in the abnormal taping skin sensitivity test within the last 30 days.

### 2.2. Sample Size Calculation

This study calculated the number of adult female softball players required using G*Power 3.1.9.2 (G* Power 3.1.9.2, Heinrich–Heine–Universität, Düsseldorf, Germany). A pilot study was conducted on 10 adult female softball players. The effect size variable was the Rectus femoris muscle activity. The input parameters were group 1 (mean: 18.25 ± 6.91) and group 2 (mean: 11.32 ± 6.78). Thus, the calculation suggested that we required 34 study participants, 17 in each group, where the effects size d was 1.0123, alpha error was 0.05, and power (1-β err prob) was 0.8. Thus, we aimed to enroll 40 participants assuming that the dropout rate would be 20%.

### 2.3. Procedure

The participants were randomly assigned to KTSE group and SKTSE group using a Randomizer (https://www.randomizer.org, accessed on 10 October 2021). Three physical therapists with >5 years of clinical experience and a master’s degree performed the intervention and measurement. Three evaluators were selected based on the following conditions: they were required have (1) been a physical therapist for >5 years and (2) received training on performing electromyography (EMG), Myoton, and strength-measuring instruments. To maintain the accuracy and consistency of measurement, one therapist was dedicated to one measurement. All participants were measured before and 6 weeks after the start of the experiment for lower-extremity muscle activity, lower-extremity muscle strength, muscle tone, and dynamic stability. The participants were randomly assigned to a KTSE group and a SKTSE group using a randomizer. During the 6-week study period, 2 participants from the KTSE group and two from the SKTSE group did not receive the assigned intervention. Thus, 40 participants finally participated in this study (Figure 1).

### 2.4. Intervention

The KTSE group performed KT with SE. The KTSE group was underwent training for 30 min/day, three times a week, for 6 weeks, i.e., a total of 18 times throughout the study. The KTSE group used the original KT product (Kinesio Holding Corporation, Albuquerque, NM, US), The method was also modified and supplemented [14,15]. The activity involved the following steps: (1) vastus medialis: the participant sat in the sitting position. KT was attached using a tension of 10–15% on a straight line passing the pes anserinus from the inner surface of the patella. However, the direction was from above to the medial side of the patella; (2) vastus lateralis: the participant sat in the sitting position. KT was performed from the great trochanter of the femur to the outside of the patella, and the tape was attached using a tension of 10–15%; and (3) rectus femoris: the participant sat in the sitting position. KT was applied from the base of the anterior inferior iliac spine to the center of the patella using a tension of 10–15% (Figure 2). In brief, the members of the KTSE group performed the following activity: keeping their arms crossed, with both feet wider than the shoulder width and the toes in a comfortable position, they performed 10 × 5 sets using the Timer Plus application at a speed of 5 s in the sitting position; a 1 min rest was allowed between sets (Figure 3).

The SKTSE group performed sham KT with SE. The SKTSE group performed the activity for 30 min/day, three times a week, for 6 weeks, i.e., a total of 18 times throughout the study period. In the SKTSE group, 5 cm of the original Kinesio Tape (Kinesio Holding Corporation, Albuquerque, NM, US) was attached horizontally on the knee bone to the far side of the quadriceps muscles to cut the tape into an I-shape and split it straight into the I-shape. SE was performed similarly as in the KTSE group (Figure 4).

KT was applied to both groups every 48 h by modifying and securing research results such as the rim [16] and for 48–72 h after the squat exercise, and they were not allowed to participate in any extra training, except for normal training in sports.

### 2.5. Measurement Variables

#### 2.5.1. Muscle Activity in the Lower Extremities

Muscle activity of the lower extremities was also measured by using EMG Noraxon Mini DTS (Noraxon, Inc., Scottsdale, AZ, USA). The common mode rejection ratio (CMRR) was 100 dB or more, and the input impedance was 100 Mohm or more. EMG signals were collected at a sampling rate of 1500 Hz with the notch filter set at 60 Hz. The EMG signal was obtained from 10 to 500 Hz using MyoResearch-XPME 1.07 (Noraxon, Inc., Scottsdale, AZ, USA) software. The first order high-pass and fourth order low-pass were filtered using Butterworth filters. The collected data were converted into root mean square values for analysis. To analyze the isometric contraction phase—which occurred during the 5 s-long pause following the execution of motion—only the data from the middle 3 s were used, with the exclusion of the initial and final second. To reduce the skin impedance of subjects, alcohol prep pads were applied on the point of attachment of the electrodes before evaluation. The point of attachment of the electrodes for the vastus medialis muscle was 4 cm above the medial surface of the patella, 3 cm on the medial side, 55° from the long axis of the patella, and the vastus lateralis muscle was 10–15 cm above the top surface of the patella, with a position of 15° from the long axis of the patella at a height of 6–8 cm. The flat-legged root was attached two-thirds of the way along the straight line between the knee bone and the lower front part [17]. The measurement posture measured three full squats and three half squats in the same posture as the squat exercise.

#### 2.5.2. Muscle Strength Test of the Lower Extremities

Muscle strength was measured using a Commander Muscle Tester from JTech Medical (UT, US), a digital muscle dynamometer. The digital muscular system guarantees 99% accuracy, and the reliability within the measurer and between the measurers is r = 0.90–0.96 and r = 0.76–0.97, respectively [18]. At the time of measurement, the participants bent the knee to 90° in the sitting position for knee joint measurement and moved parallel to a line on the ground to avoid effects of gravity in a straight lying position for ankle joint measurement [19]. Each measurement was performed three times. A break of 15 s was allowed after every 10 s of measurement to avoid muscle fatigue, followed by the final measurement.

#### 2.5.3. Muscle Tension in the Lower Extremities

Muscle tension was measured using MyotonPRO. For the measurement, the number of times the tap was repeated in the multi-scan mode was five times, the tap time (mechanical impulse transfer time) was 15 milliseconds, and the transfer interval was 8 s. The measurement position was attached to the long axis of the knee bone, 4 cm above the inner wide muscle knee bone and 55° above the inner wide muscle knee bone simultaneously as the electrode was attached [20,21]. Before the measurement, the participants were allowed to lie down in the right posture comfortably, and during the measurement, the participants were instructed to avoid unnecessary movements

#### 2.5.4. Dynamic Stability of the Lower Extremities (Side Bop Test)

Side hop tests had a high degree of reliability (ICC = 0.84). The method involved test intervals of 30 cm and returning to the start position, and the round trip time of 10 round trips was measured in units of 0.01 s. If the participant stepped on the line during the measurement, it was considered a failure if one did not cross the given distance; the test was repeated after a break [22,23].

### 2.6. Statistical Analysis

All statistical analyzes in this study were performed using IBM SPSS version 20.0, (IBM Corp., Armonk, NY, USA). The normality test was analyzed with the Shapiro–Wilk test. The variable of the dominant leg was assessed with chi-square test, while height, weight, career, and the homogeneity of the dependent variable pretest were assessed with an independent sample *t*-test. The effects of intervention muscle activity, muscle tone, muscle power, and dynamic stability test were performed using the two-way repeated measures ANOVA analysis. The pre- and post-tests were the time (within-participant factors) results. The group-by-time (between-participant factors) were the KTSE and SKTSE group results. When significant differences were observed in group-by-time (main effects or interactions) analyses, the independent test (Time * Groups) and paired *t*-test (Time) were used in post hoc analysis. All statistical significance levels of date were set at α = 0.05.

## 3. Results

### 3.1. General Characteristics of Participants

Table 1 shows the general characteristics of participants (Table 1).

### 3.2. Comparison of Muscle Activity of Lower Extremities between the Groups

Significant between-participant changes were observed for vastus medialis muscle activity (VMMA) (F = 68.195, *p* = 0.000) and vastus lateralis muscle activity (VLMA) (F = 185.102, *p* = 0.000). The VMMA and VLMA showed a significantly greater increase in the KTSE group than in the SKTSE group. Significant within-participant changes were observed for VMMA (F = 12.108, *p* = 0.001) and VLMA (F = 9.814, *p* = 0.003). Significant between-participant changes were observed for rectus femoris muscle activity (RFMA) (F = 111.397, *p* = 0.000). The RFMA showed a significantly greater increase in the KTSE group than in the SKTSE group. Significant within-participant changes were observed for RFMA (F = 11.958, *p* = 0.001) (Table 2).

### 3.3. Comparison of Muscle Tone of Lower Extremities between the Two Groups

Significant between-participant changes were observed for vastus medialis muscle tone (VMMT) (F = 36.006, *p* = 0.000) and vastus lateralis muscle tone (VLMT) (F = 48.670, *p* = 0.000). The VMMT and VLMT showed a significantly greater increase in the KTSE group than in the SKTSE group. Significant within-participant changes were observed for VMMT (F = 4.766, *p* = 0.035) and VLMT (F = 5.454, *p* = 0.025). Significant between-participant changes were observed for rectus femoris muscle tone (RFMT) (F = 54.852, *p* = 0.000). The RFMT showed a significantly greater increase in the KTSE group than in the SKTSE group. Significant within-participant changes were observed for RFMT (F = 4.673, *p* = 0.037) (Table 3).

### 3.4. Comparison of Muscle Power of Lower Extremities between the Two Groups

Significant between-participant changes were observed for quadricep muscle power (QMP) (F = 81.572, *p* = 0.000) and soleus muscle power (SMP) (F = 150.479, *p* = 0.000). The QMP and SMP showed a significantly greater increase in the KTSE group than in the SKTSE group. Significant within-participant changes were observed for QMP (F = 4.645, *p* = 0.038) and SMP (F = 4.616, *p* = 0.038) (Table 4).

### 3.5. Comparison of Side Hop Test between the Two Groups

Significant between-participant changes were observed for the side hop test (SHT) (F = 171.904, *p* = 0.000). The SHT showed a significantly greater increase in the KTSE group than in the SKTSE group. Significant within-participant changes were observed for SHT (F = 33.207, *p* = 0.000) (Table 5).

## 4. Discussion

Since the knee performs motions by interacting with the joints of other lower extremities, it is difficult to move it alone in biomechanical function [24]. Thus, management of the knee is thus more important than anything else, and has a great impact on the performance of softball players, as softball involves knee movement to produce momentary acceleration. Therefore, this study focused on the effects of KT with squat exercise on lower-extremity muscle activity, muscle strength, muscle tension, and dynamic stability in softball players. Participants were grouped into a KTSE group that performed KT on the quadricep muscles and a SKTSE group that attached KT from the knee bone to the far side of just above the quadricep muscles. In a previous study, KT was attached to the quadricep muscle, the primary muscle for moving the knee joint. The results of KTSE, open-chain exercise, and cross-chain exercise squats for 10 knee-joint patients showed that the activity of the broad medial muscle was significantly higher in the cross-chain exercise [25], which was selected based on previous studies.

The KTSE group showed a statistically significant difference. This indicates that KTSE increases the muscle activity and dynamic stability of the lower extremities of athletes and reduces muscle tension better than squat exercises with sham KT. This was performed in parallel with KT for 6 weeks on 24 patients with knee joint-related diseases, which increased the muscle strength and stability of the knee joint and effectively improved the intrinsic capacity. This shows a significant increase in the activity of the quadricep muscle and the stability of the knee joint while improving the proprioception [26]. This is consistent with the result of a study wherein taping was applied to a medial patella glide in patients with patellofemoral pain syndrome; the muscle activity of the medial scapularis and lateral gluteus maximus significantly increased in their study [27].

In addition, a study on athletes found that KT is more effective in terms of muscle strength and jump performance than knee guards, and that therapists can apply KT to participants who underwent treatment and rehabilitation to support knee muscle tissues and expect improved knee muscle performance during healing [28]. These insights from the previous study support our results, i.e., squat exercises combined with KT of the lower extremities showed significant differences in muscle activity, muscle strength, and dynamic stability.

Both groups showed improvements in muscle activity, muscle strength, dynamic stability, and muscle tension relief in this study. However, there was a significant difference between the KTSE group and the SKTSE group. This suggests that the additional effect in the KTSE group was due to KT. This additional effect can be attributed to the structural changes caused due to the additional physical space provided by lifting the skin and relieving the pressure of the receptors involved in pain located under the dermis.

The significant difference in the KTSE group could be attributed to the spatial summation (additive effect) effect resulting from the simultaneous depolarization of a large number of presynaptic nerve fibers due to increased muscle tension by KT. This improvement in muscle strength may be attributed to a higher level of tension in the muscles from the combined effect. As for the second mechanism, intensifying the muscle stimulation using KT may have increased contractility, which is a natural response from muscles [29]. Finally, reciprocal innervation may also occur at the level of the spinal cord, starting at sensory receptors in the skin [30].

Although unclear, the effectiveness of KT has been confirmed in many studies. Our findings are consistent with those of many studies. KT is still used in many studies, and its effectiveness has been proven repeatedly. As a result, it is widely applied in various fields such as the rehabilitation and training of athletes and the prevention of injuries, and numerous studies recommend the use of KT. If certain variables—length of the gastrocnemius muscle and flexibility of the pelvis and hip joints—were determined in the current study, future research could utilize both KT and SE simultaneously, regardless of participant ages. However, we recommend monitoring and correcting the following points before engaging in exercises: whether the participants are allergic to the tape used for KT and whether the participants experience pain while executing the squat exercise motions.

There are a few limitations to this study. Muscle fatigue of the lower extremity was not measured, the effects of KT were not verified, and since there were no comparisons made with placebo taping, the placebo effects of each taping could not be verified. Considering the small sample size, it is difficult to generalize the results to the general population. In addition, a longitudinal study involving simultaneous squat exercise accompanied by KT could not be conducted. Therefore, future research should address these limitations to prove the effects of KT and investigate the differences in a larger number of participants

## 5. Conclusions

The following conclusions could be drawn from this study by comparing the effects of KTSE on the lower extremity muscle activity, muscle strength, muscle tension, and dynamic stability of female softball players. KTSE had a more significant effect on lower-extremity muscle activity, muscle strength, muscle tension, and dynamic stability than SKTSE. Therefore, we believe that KTSE can be considered as a program to increase lower-extremity function when planning a training program for softball players.

## Figures and Tables

**Figure 1 ijerph-19-00276-f001:**
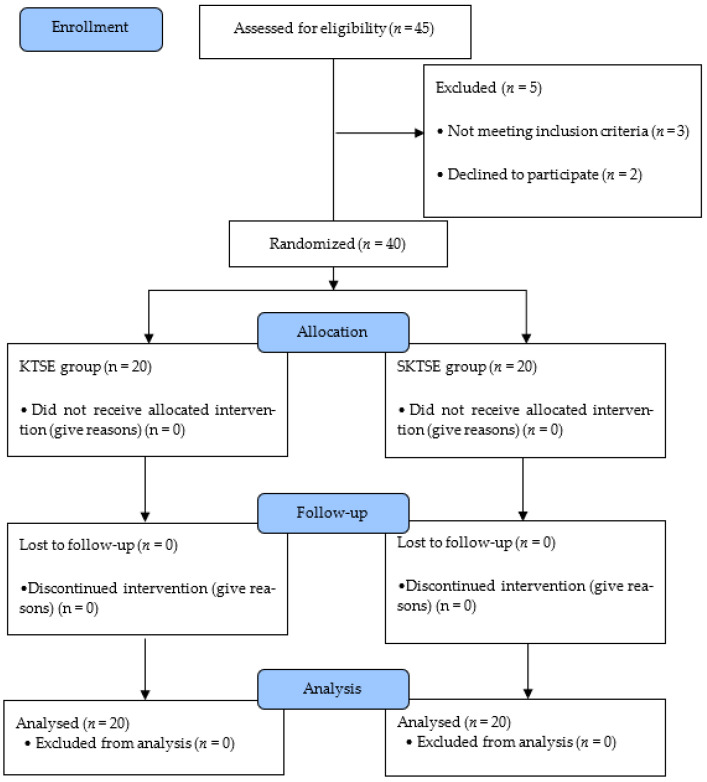
Flow diagram of the study.

**Figure 2 ijerph-19-00276-f002:**
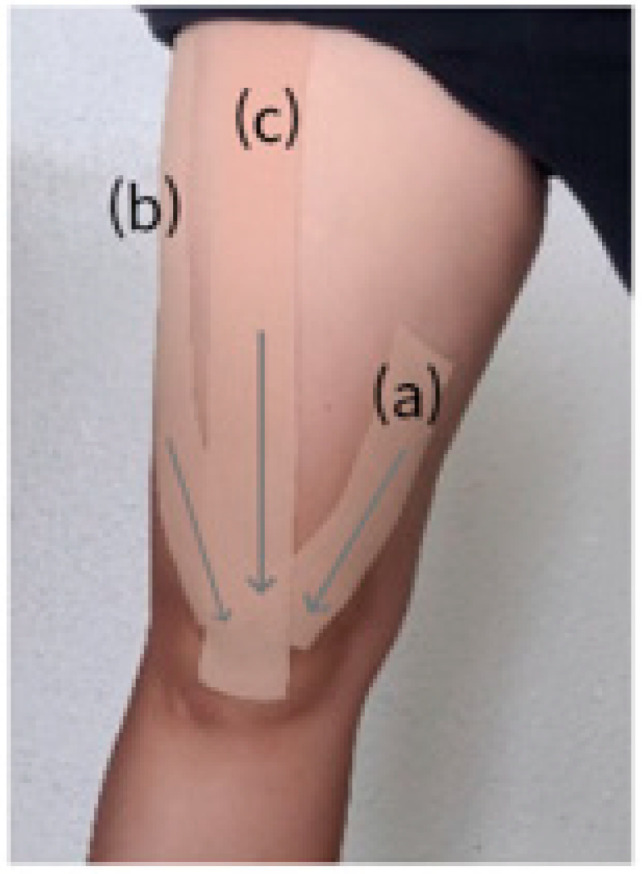
Kinesio taping technique. (a) vastus madialis taping; (b) vastus lateralis taping; (c) rectus femoris taping.

**Figure 3 ijerph-19-00276-f003:**
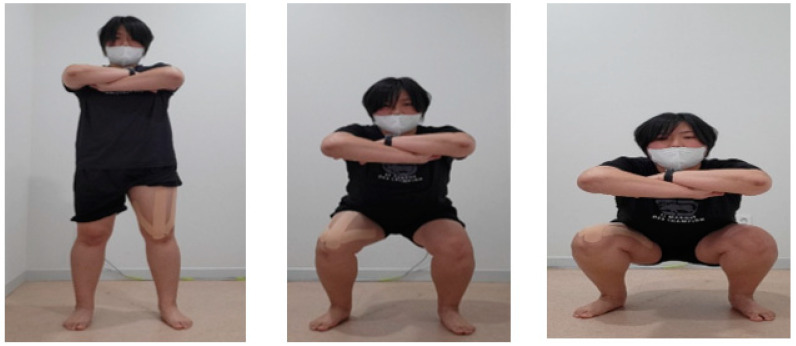
Kinesio taping with squat exercise.

**Figure 4 ijerph-19-00276-f004:**
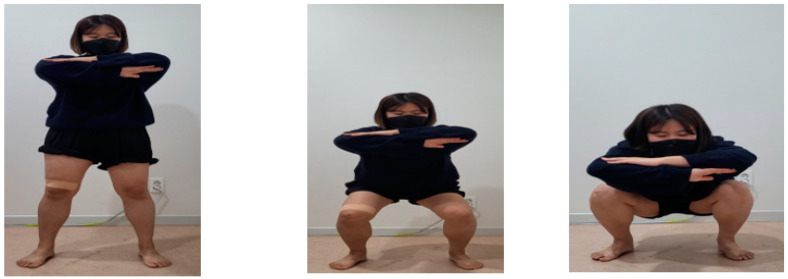
Sham Kinesio taping with squat exercise.

**Table 1 ijerph-19-00276-t001:** General and clinical characteristics of subjects.

Variable	KTSE Group (*n* = 20)	SKTSE Group (*n* = 20)	*p* ^b^	*p* ^c^
Height (cm) ^a^	165.70 ± 7.41	164.20 ± 4.71	0.449	
Weight (kg) ^a^	68.34 ± 10.54	70.30 ± 12.27	0.592	
Age (year) ^a^	21.85 ± 1.53	21.95 ± 1.90	0.856	
Career (year) ^a^	7.60 ± 1.54	7.30 ± 1.75	0.568	
Dominant leg (right/left)	17/3	16/4		0.677

^a^ Mean ± SD. ^b^ Independent *t*-test, ^c^ Chi-square test. KTSE group, Kinesio taping with squatting exercise group, SKTSE group, sham Kinesio taping with squatting exercise group.

**Table 2 ijerph-19-00276-t002:** Comparison of muscle activity between the KTSE and SKTSE groups.

	KTSE Group (*n* = 20)	SKTSE Group (*n* = 20)	*ES*	*F*	*p*
Vastus medialis muscle activity (%RVC)					
Pretest	181.34 ± 18.82 ^a^	180.46 ± 19.84	0.0455	12.018	0.001 *^,1^
Posttest	222.55 ± 25.71 ^b,c^	197.23 ± 22.57 ^b^	1.0467		
Vastus lateralis muscle activity (%RVC)					
Pretest	194.84 ± 23.66	186.04 ± 26.49	0.3504	9.814	0.003 *^,1^
Posttest	220.05 ± 25.66 ^b,c^	202.74 ± 24.60 ^b^	0.6887		
Rectus femoris muscle activity (%RVC)					
Pretest	195.97 ± 21.55	191.14 ± 22.92	0.2171	11.958	0.001 *^,1^
Posttest	224.55 ± 22.56 ^b,c^	205.61 ± 22.93 ^b^	0.7887		

^a^ Mean ± SD, * *p* < 0.05, ^b^ Significant differences between pre- and post-test (*p* < 0.05). ^c^ The KTSE group improved more than the SKTSE group. ^1^ Analyzed by two-way repeated measures ANOVA. KTSE group, Kinesio taping with squatting exercise group, SKTSE group, sham Kinesio taping with squatting exercise group, ES: effect sizes d.

**Table 3 ijerph-19-00276-t003:** Comparison of muscle tone between the KTSE and SKTSE groups.

	KTSE Group (*n* = 20)	SKTSE Group (*n* = 20)	*ES*	*F*	*p*
Vastus medialis muscle tone (Hz)					
Pretest	12.31 ± 0.88 ^a^	12.07 ± 0.61	0.3170	4.766	0.035 *^,1^
Posttest	12.62 ± 0.73 ^b,c^	12.22 ± 0.63 ^b^	0.5867		
Vastus lateralis muscle tone (Hz)					
Pretest	13.11 ± 0.73	12.74 ± 0.81	0.4799	5.454	0.025 *^,1^
Posttest	13.73 ± 0.86 ^b,c^	13.04 ± 0.68 ^b^	0.8900		
Rectus femoris muscle tone (Hz)					
Pretest	13.99 ± 0.62	13.65 ± 0.92	0.4334	4.673	0.037 *^,1^
Posttest	14.42 ± 0.59 ^b,c^	13.88 ± 0.91 ^b^	0.7041		

^a^ Mean ± SD, * *p* < 0.05, ^b^ Significant differences between pre- and post-test (*p* < 0.05). ^c^ The KTSE group improved more than the SKTSE group. ^1^ Analyzed by two-way repeated measures ANOVA. KTSE group, Kinesio taping with squatting exercise group, SKTSE group, sham Kinesio taping with squatting exercise group, ES: effect sizes d.

**Table 4 ijerph-19-00276-t004:** Comparison of muscle power between the KTSE and SKTSE groups.

	KTSE Group (*n* = 20)	SKTSE Group (*n* = 20)	*ES*	*F*	*p*
Quadriceps muscle power (N)					
Pretest	93.04 ± 6.96 ^a^	92.01 ± 11.62	0.1075	4.645	0.038 *^,1^
Posttest	106.52 ± 6.69 ^b,c^	100.30 ± 9.01 ^b^	0.7838		
Soleus muscle power (N)					
Pretest	165.72 ± 16.47	164.40 ± 15.59	0.0823	4.616	0.038 *^,1^
Posttest	182.54 ± 13.78 ^b,c^	176.20 ± 13.89 ^b^	0.4583		

^a^ Mean ± SD, * *p* < 0.05, ^b^ Significant differences between pre- and post-test (*p* < 0.05). ^c^ The KTSE group improved more than the SKTSE group. ^1^ Analyzed by two-way repeated measures ANOVA, KTSE group, Kinesio taping with squatting exercise group, SKTSE group, sham Kinesio taping with squatting exercise group, ES: effect sizes d.

**Table 5 ijerph-19-00276-t005:** Comparison of side hop test between the KTSE and SKTSE groups.

	KTSE Group (*n* = 20)	SKTSE Group (*n* = 20)	*ES*	*F*	*p*
Side hop test (s)					
Pretest	10.50 ± 1.72 ^a^	10.79 ± 2.17	0.1481	33.207	0.000 *^,1^
Posttest	9.02 ± 1.61 ^b,c^	10.21 ± 2.16 ^b^	0.6247		

^a^ Mean ± SD, * *p* < 0.05, ^b^ Significant differences between pre- and post-test (*p* < 0.05). ^c^ The KTSE group improved more than the SKTSE group. ^1^ Analyzed by two-way repeated measures ANOVA. KTSE group, Kinesio taping with squatting exercise group, SKTSE group, sham Kinesio taping with squatting exercise group, ES: effect sizes d.

## Data Availability

Not applicable.

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
