# Peer review of "Effects of Kinesio Taping with Squat Exercise on the Muscle Activity, Muscle Strength, Muscle Tension, and Dynamic Stability of Softball Players in the Lower Extremities: A Randomized Controlled Study"

_ijerph, 2021, doi:10.3390/ijerph19010276_

Round 1

Reviewer 1 Report

In this study the authors have studied the effect of kinesio tapping with squatting exercise on the lowe limb muscle activity. The authors have shown the positive effect of kinesio tapping on the lower limb's muscle function.  The study is mostly complete. However, he authors didnot discuss the limitation of their study in the discussion section which can strengthen the manuscript. My comments are provided below

  1. In Table 2, their is still a change between pretest and posttest subgroup  in PKTSE group. The authors should mention the statistical significance of it.
  2. The authors should provide the reason for not having a group with only kinesio tapping in their study.
  3. The study has been conducted only for six weeks. The authors should discuss whether the long term study will be more beneficial or not in this scenario.
  4. The authors should discuss whether the effect of kinesio tapping with squatting exercise is beneficial to other age group as the median age of the study is 21 years.

Reviewer 2 Report

The manuscript takes up an interesting research problem, but unfortunately it is illegible and incomprehensible. The language side should be corrected. There are incomprehensible whole fragments (marked in blue) and editorial errors (marked in yellow). The language side of the article and the mental abbreviations used make it impossible to understand. In this version, I am not able to fully evaluate this study.

Reviewer 3 Report

Comments for “Effects of Kinesio Taping with Squat Exercise on the Muscle Activity, Muscle Strength, Muscle Tension, and Dynamic Stability of Softball Players in the Lower Extremities: a randomized controlled study”

The purpose of this study was to examine the combination effect of kinesiology tape (KT) and squat exercise on lower body muscle EMG activity, muscle strength, muscle tone of the lower extremities, and dynamic stability in healthy female softball players. Forty softball players were randomly assigned into the combination group (KTSE) and the placebo group (PKTSE), and both groups were trained for a total of 18 times (3 times per week for 6 weeks). It was reported that both groups improved all the dependent variables, but KTSE showed greater improvements in muscle activity, muscle strength, muscle tone, and dynamic stability than the PKTSE group.

Unfortunately, there is too much information missing from the paper. Additionally, many contents within the paper were not consistent. Following are my comments:

First of all, the writing needs to be improved. There are lots of wording in sentences should be rephrased. The authors are encouraged to find a language editing service to polish the paper.

Regarding the participants, please provide more information regarding the level of the athletes. Based on the description, it seems that the authors were focusing on female players. But in the inclusion criteria, it seems that players (male and female) under 30 years of age were included. Additionally, the abstract indicated that there were 50 subjects, but in the main text there were 40.

The figures were all misplaced. There was no Figure 1 content (experimental flow) in the main text. In Figure “Placebo knee kinsio taping”, what is the (d)? For kinesio taping figure, the pictures were taken from the side view, which doesn’t show where exactly the tapes were placed. For placebo group, I don’t think the figure is necessary.

For the placebo group, I don’t believe the taping was conducted correctly. Putting a tape horizontally above the knee is not a placebo. A true placebo should mimic every aspect of the experimental group (putting the tapes on the target muscles exactly the same as the KTSE), but with zero/no tension.

For muscle activity measurement, it is unknown which muscle(s) were tested based on the description. The paper also misses some EMG data acquisition and analysis information: EMG sampling frequency, filtering, and EMG normalization. Similar for other measurements, the details are needed so others can replicate the experiment.

Where is the Statistical analysis section???

Round 2

Reviewer 1 Report

The authors have addressed all my concerns by providing necessary justification and including the limitations of the study. I  support the publication of the manuscript.

Author Response

Thank you for your understanding and support for the publication of the manuscript. 

Reviewer 2 Report

The revised manuscript is legible, requiring minor editorial correction (highlighted in yellow).

Reviewer 3 Report

Thank you for revising the manuscript. I still have some questions/concerns:

Figure 1. The pdf is missing a, b, and c. In fact, you could delete 1d, because your Figure 3 already showed the sham condition.

Figure 4 cannot be found in the main text. I believe the location should be before the taping figures.

Regarding 2.5.1. EMG, you only answered a portion of the previous question. But what about the filtering frequency (high and low pass)? Did you have the EMG normalization process for the amplitude?  

Statistical analysis was not done correctly. You have three groups with pre and post testing. So, you need to conduct a mixed-factorial ANOVA (group x time) to examine your dependent variables.
